# Ice velocity of Jakobshavn Isbræ, Petermann Glacier, Nioghalvfjerdsfjorden and Zachariæ Isstrøm, 2015-2017, from Sentinel 1-a/b SAR imagery

Adriano Lemos[1], Andrew Shepherd[1], Malcolm McMillan[1], Anna E. Hogg[1], Emma Hatton[1], Ian Joughin[2]

[1]Centre for Polar Observation and Modelling, University of Leeds, Leeds, United Kingdom
[2]Polar Science Center, Applied Physics Laboratory, University of Washington, Seattle, Washington, USA

*Correspondence to*: Adriano Lemos (A.G.Lemos14@leeds.ac.uk)

## Abstract

Systematically monitoring Greenland's outlet glaciers is central to understanding the timescales over which their flow and sea level contributions evolve. In this study we use data from the new Sentinel-1a/b satellite constellation to generate 187 velocity maps, covering 4 key outlet glaciers in Greenland; Jakobshavn Isbræ, Petermann Glacier, Nioghalvfjerdsfjorden and Zachariæ Isstrøm. These data provide a new high temporal resolution record (6 days averaged solutions) of each glacier's evolution since 2014, and resolve recent seasonal speedup periods and inter-annual changes in Greenland outlet glacier speed with an estimated certainty of 10 %. We find that since 2012, Jakobshavn Isbræ has been decelerating, and now flows approximately 1250 m yr$^{-1}$ (10 %) slower than 5 years previously, thus reversing an increasing trend in ice velocity that has persisted during the last decade. Despite this, we show that seasonal variability in ice velocity remains significant; up to 750 m yr$^{-1}$ (14 %) at a distance of 12 km inland of the terminus. We also use our new dataset to estimate the duration of speedup periods (80-95 days), and to demonstrate a strong relationship between ice front position and ice flow at Jakobshavn Isbræ, with increases in speed of ~1800 m yr$^{-1}$ in response to 1 km of retreat. Elsewhere, we record significant seasonal changes in flow of up to 25 % (2015) and 18 % (2016) at Petermann Glacier and Zachariæ Isstrøm, respectively. This study provides a first demonstration of the capacity of a new era of operational radar satellites to provide frequent, and timely, monitoring of ice sheet flow, and to better resolve the timescales over which glacier dynamics evolve.

## 1. Introduction

Between 1992 and 2011, the Greenland Ice Sheet lost mass at an average rate of 142±49 Gt yr$^{-1}$ [Shepherd et al., 2012], increasing to 269±51 Gt yr$^{-1}$ between 2011 and 2014 [McMillan et al., 2016]. Ice sheet mass balance is determined from the surface mass balance and ice discharge exported from the ice sheet [van den Broeke et al., 2009]. In 2005, dynamic imbalance was responsible for roughly two-thirds of Greenland's total mass balance, making an important contribution to freshwater input into the ocean and 0.34 mm yr$^{-1}$ to the global sea level rise at that time [Rignot & Kanagaratnam, 2006]. Despite the anomalous atmospheric warming events, especially in 2012 [Tedesco et al., 2013], presenting a more spatially extensive and longer lasting surface melt during this period, marine-terminating outlet glaciers in Greenland still contributed with roughly 30 % (2000–2012) of total mass loss [Enderlin et al., 2014]. The observed acceleration of many marine-based glaciers in the western and northern regions of Greenland over the last decade may have been driven by rises in air and adjacent ocean temperatures, which enhanced the surface melting and terminus retreat [Holland et al., 2008; Moon et al, 2014; Moon et al., 2015]. The associated increases in basal sliding and calving of their ice fronts in turn produce enhanced discharge, leading to dynamical imbalance and additional ice loss [Joughin et al., 2010; Joughin et al., 2014]. Acceleration of marine-terminating glaciers is, however, highly variable in space and time [Howat et al., 2010; Moon et al., 2012; Enderlin et al., 2014], due to the geometry of individual glaciers (Felikson et al., 2017), and the high spatial variability in the forcing mechanisms (Jensen et al., 2016; Carr et al., 2017). This complexity in glacier response challenges efforts to model their

future evolution [Joughin et al., 2012; Bondzio et al., 2017] and, therefore, frequent and systematic monitoring is essential to understand the processes governing their dynamic stability and contribution so future mean sea level rise [Joughin et al., 2010; Shepherd et al., 2012].

Ice motion measurements are essential for monitoring ice sheet dynamics and ice discharge, and for assessing an ice sheet's mass budget [Joughin et al., 1995]. At present, the only way to monitor ice velocity at a continental scale is through satellite imagery. Glacier velocities were first measured using Landsat satellite data acquired during the 1970s through digital optical image comparison [Lucchitta & Ferguson, 1986]. Currently, optical images are still largely used for mapping glaciers velocity at large scale (e.g. Dehecq et al., 2015; Fahnestock et al., 2016; Armstrong et al., 2017). However, due to the dependency upon daylight
conditions and the limited acquisitions across the polar Regions, the use of Synthetic Aperture Radar (SAR) images has become common since the launch of ERS-1 in 1991. In the following decades, these data have been used to monitor dynamic processes occurring across remote areas such as the Greenland and Antarctic ice sheets [Joughin et al., 2010; Rignot & Mouginot, 2012; Nagler et al., 2015 Mouginot et al., 2017]. More recently, after the launch by the European Space Agency (ESA) of the Sentinel 1-a and 1-b satellites, in April 2014 and April 2016 respectively, many key ice margin areas are systematically monitored every 6 to
12 days. This novel dataset provides the opportunity to systematically monitor the dynamical process driving glacier ice velocity over periodic and short temporal scales. Here we use the Sentinel SAR archive to investigate the temporal variation in ice flow since October 2014 at four large outlet glaciers of the Greenland ice sheet.

## 2.    Study areas

In this study, we map ice velocity of the Jakobshavn Isbræ (JI), Petermann Glacier (PG), Nioghalvfjerdsfjorden (79-G) and Zachariæ Isstrøm (ZI), which are four of the largest marine-based ice streams in Greenland. Combined they contain ice equivalent to 1.8 m of global sea-level rise [Mouginot et al., 2015; Jensen et al., 2016], and drain ~21.5 % of Greenland's ice [Rignot & Kanagaratnam, 2006; Rignot & Mouginot, 2012; Münchow et al, 2014].

Jakobshavn Isbræ terminates in the Ilulissat Icefjord in western Greenland (Figure 1a), and is the fastest glacier draining the ice sheet [Enderlin et al., 2014; Joughin et al., 2014]. During the late 1990s, the ice tongue experienced successive break up events and the glacier began to speedup, exhibiting annual increases in speed of 7 % per year from 2004 and 2007 [Joughin et al., 2008a; Joughin et al., 2012; Joughin et al., 2014]. Until 2012 and 2013, the speed up has continued, reaching maximum velocities in excess
of 17 km yr$^{-1}$ [Joughin et al., 2012; Joughin et al., 2014]. It has been suggested [van de Wal et al., 2015] that the speedup over this period in the southwest of Greenland might enhanced by anomalously high melting across the ice sheet surface [Tedesco et al., 2013]. Jakobshavn Isbræ is susceptible to changes in the adjacent ocean and Holland et al. [2008] have shown that warm water originating in the Irminger Sea likely enhanced basal melting and weakened the floating ice tongue, triggering its break up in 1997. Furthermore, Gladish et al. [2015] showed that subsequent changes which, occurred between 2001–2014, were mainly triggered by
changes in Ilulissat Icefjord water temperatures adjacent to the glacier. At present, JI is a tidewater glacier and has a bimodal behaviour, retreating by ~3 km during summer and advancing by a similar amount during winter seasons [Cassotto et al., 2016]. Moreover, as showed by Jensen et al. [2016] through analysis of optical images from 1999 to 2013, it has not exhibited an unusually large change in area (-10.3 km$^2$ yr$^{-1}$).

Petermann Glacier flows into the Hall Basin in the Nares Strait in Northwest Greenland (Figure 1b), and has a perennial floating ice tongue of 1280 km$^2$ in area [Hogg et al, 2016]. PG is grounded on bedrock ~300 m below sea level and, therefore, is also influenced

by the adjacent ocean [Münchow et al, 2014; Hogg et al., 2016]. The retreat of the ice stream calving front led to an area decrease of 352 km$^2$ from 1959 to 2008, 270 km$^2$ in 2010 and 130 km$^2$ in 2012 [Johannessen et al., 2013]. It is considered a dynamically stable marine-terminating glacier despite several grounding line advancing and retreating events between 1992 and 2011, with a terminus retreat rate of 25.2 m a$^{-1}$ [Hogg et al., 2016]. PG has an average velocity of ~1100 m yr$^{-1}$ at its grounding line since the 1990s [Rignot, 1996; Rignot & Steffen, 2008] and a multi-annual trend (2006–2010) in flow speed of 30 m yr$^{-2}$ [Nick et al., 2012]. The ice shelf is thicker than 100 m and it is 15 km wide, with low resistive stresses along flow due to the limited attachment to the fjord walls, diminishing the velocity response after calving events [Nick et al., 2012].

Nioghalvfjerdsfjorden and Zachariæ Isstrøm are situated in the northeast of Greenland (Figure 1c and Figure 1d respectively). The two glaciers together drain more than 10 % of the Greenland Ice Sheet [Rignot & Mouginot, 2012], and their maximum velocities are found near the grounding line. They have exhibited different behaviour in recent years, although located in the same region. 79-G underwent a modest velocity increase of ~150 m yr$^{-1}$ between 2001 and 2011 at the grounding line [Khan et al., 2014]. In contrast, during the same period, ZI exhibited a much larger increase in speed greater than 600 m yr$^{-1}$ [Khan et al., 2014]. The ice thinning rates above the grounding line varies from 5.1 m yr$^{-1}$ in ZI (2010–2014) to 1.4 m yr$^{-1}$ in 79-G (2012–2014) [Mouginot et al., 2015]. Between 1999 and 2013, ZI has undergone an average area change of -26.0 km$^2$ yr$^{-1}$, due to break off of the ice tongue and is now a tidewater glacier [Khan et al., 2014; Jensen et al., 2016]. In contrast, 79-G had a much lower average area change during the same period of -4.7 km$^2$ yr$^{-1}$ and still retains a small ice shelf [Jensen et al., 2016], although recent ice shelf thinning [Mouginot et al., 2015] may increase vulnerability to break up in the future.

## 3.    Data and Methodology

To map ice velocity, we used Single Look Complex (SLC) Synthetic Aperture Radar images acquired in the Interferometric Wide swath (IW) mode from the Sentinel-1a and Sentinel-1b satellites. Data used in this study were acquired in the period spanning from October 2014 to February 2017 and from October 2016 to February 2017, for Sentinel-1a and Sentinel-1b respectively (Figure S2 and Table S1). Each satellite has a repeat cycle of 12 days and 180 degrees orbital phasing difference, resulting in a revisit time of 6 days over the same area after the Sentinel-1b launch. The Sentinel SAR instruments operate at C-Band, with a centre frequency of 5.405 GHz, corresponding to a wavelength of 5.55 cm. The IW mode has a 250 km swath and spatial resolution of 5 m in ground range and 20 m in azimuth. It has burst synchronization for interferometry and acquires data in 3 sub-swaths, each containing a series of bursts, which are acquired using the Terrain Observation with Progressive Scans SAR (TOPSAR) imaging technique [Yague-Martinez et al., 2016]. We followed the workflow described below to derive 187 ice velocity maps from pairs of Sentinel-1a/b SAR images over Jakobshavn Isbræ, Petermann Glacier, Nioghalvfjerdsfjorden and Zachariæ Isstrøm, using the GAMMA-SAR software [Gamma Remote Sensing, 2016].

We used the SAR intensity tracking technique [Strozzi et al., 2002] to estimate surface ice velocities due to glacier flow, assuming that the ice flow occurs parallel to the surface. This method uses a cross correlation algorithm applied to image patches [Strozzi et al. 2002; Pritchard et al. 2005, Paul et al. 2015] to estimate offsets between similar features, such as crevasses and radar speckle patterns, in two co-registered SAR images (Table S1). Images were co-registered using the precise orbit information, available 20 days after the image acquisition, establishing a co-registration accuracy of 5 cm 3D 1-sigma [Sentinels POD team, 2013]. The elimination of the orbital offsets isolates displacement due to the glacier movement [Strozzi et al., 2002]. To estimate ice flow, we then used windows sizes of 350 pixels in ground range (~ 1.7 km) and 75 pixels in azimuth (~1.5 km) for each glacier, to produce a series of velocity maps with spatial resolution of 388 m in ground range and 320 m in azimuth.

Image matches with low certainty, defined as returning a normalised cross-correlation of less than 5 % of its maximum peak , were rejected and the results were then converted into displacement in ground range coordinates using the Greenland Ice Mapping Project (GIMP) digital elevation model (DEM) posted on a 90 m grid [Howat, 2014]. Along- and across- track displacement components were combined to determine the displacement magnitude, which was then converted to an estimate of annual velocity using the temporal baseline of each image pair. Final velocity products were posted on 100 m by 100 m grids. Post-processing of ice velocity data reduces noise and removes outliers [Paul et al., 2015], so we applied a low-pass filter (moving mean) twice to the data, using a kernel of 1 km by 1 km, and we reject values where the deviation between the unfiltered and filtered velocity magnitude exceeds 30 %. We apply a labelling algorithm, based on the image histogram, to identify and classify regions with similar values, excluding isolated pixels with a non-coherent area of velocity values, or where the area of the classified region was smaller than 1/1000[th] of the processed image size.

Errors in our velocity estimates arise primarily through inexact co-registration of the SAR images, uncertainties in the digital elevation model used in the terrain correction, and fluctuations in ionospheric activity and tropospheric water vapour [Nagler et al., 2015; Hogg et al., 2017]. To estimate the accuracy of our Sentinel-1 average velocity data (Figure 1 and Figure 3) we computed pixel-by-pixel errors based on the signal to noise ratio (SNR) of the cross correlation function [Hogg et al., 2017]. The SNR is the ratio between the cross-correlation function peak ($C_p$) and the average correlation level ($C_l$) on the tracking window used to estimate the velocities (Lange et al., 2007). We then averaged these estimates across all images in our temporal stack to determine the percentage errors associated with our mean velocity maps (Figure 3). Although in isolated areas the error exceeds 30 %, the mean error across the whole imaged area were approximately 10 % for JI, 7 % for PG, and 8 % for 79G and ZI. Due to the non-uniform flow, lack of stable features and remaining geometry distortions, the four glaciers exhibit higher errors across their faster flowing and steeper areas, and along the shear margins. Where localised rates of surface elevation change are high, the surface slope may have evolved away from that of the GIMP DEM used in our processing. To assess the sensitivity of our velocity estimates to this effect, we selected the JI site where thinning is most pronounced, and used airborne estimates of elevation change from IceBridge and Pre-Icebridge data acquired from the NASA Airborne Topographic Mapper (ATM) [Studinger, 2014] to update the DEM. We find that in this extreme case, the large thinning rates (~12 m yr$^{-1}$) may introduce an additional uncertainty of 200-300 m yr$^{-1}$ which may bias the velocity estimates in this region, albeit limited to the first 10 km upstream of the grounding line (Table S2). Over floating ice tongues, uncompensated vertical tidal displacement may also introduce additional uncertainty into our velocity fields. The sensitivity of our results to this effect was assessed based upon a net 50 cm tidal displacement over 6-12 day repeat period and a centre swath incidence angle of 35 degrees. We estimate that such a tidal signal would introduce ~20–40 m yr$^{-1}$ additional uncertainty into the ground range component of our velocity fields. In the context of this study, this uncertainty does not affect the results at JI or ZI, and it is limited only to the floating sections of PG and 79G.

To provide an independent evaluation of our ice velocity dataset, we finally compared them (Table S1) to independent estimates derived from TerraSAR-X (TSX) SAR imagery through the speckle tracking technique (Joughin, 2002), which has a repeat period acquisition of 11 days and spatial resolution up to 3 m [Joughin et al., 2016]. The TSX data consist of 444 image pairs covering Jakobshavn Isbræ over the period January 2009 to January 2017, 18 pairs at Petermann Glacier over the period November 2010 to December 2016, and 17 pairs at Nioghalvfjerdsfjorden over the period March 2011 to December 2016. In general, the temporal evolution of the S1-a/b measurements matches very closely with the TSX estimates. At JI, we are able to compare S1 and TSX datasets at three different locations to assess their consistency (Figure 4). Even though the flow speed at these sites is high, which typically proves more challenging for feature tracking techniques, we find good agreement between the two datasets, especially at the J1 and J2 sites, with mean differences of 40 m yr$^{-1}$ and 76 m yr$^{-1}$ respectively. However, nearer to the calving front (site Jif), the S1-a/b measurements tend to give significantly higher velocities than TSX with a mean difference of 489 m yr$^{-1}$ (5 % of the mean velocity) between the two datasets.

## 4. Results and Discussion

We used our complete Sentinel-1a/b dataset (Table S1) to generate contemporary, time-averaged velocity fields at each of our study sites (Figure 1). To investigate spatial and temporal variations in ice velocity, we then extracted profiles in the along- and across-flow directions, together with time series at fixed glacier locations (Figure 1). Our velocity profiles in Jakobshavn Isbræ, Petermann Glacier, Nioghalvfjerdsfjorden and Zachariæ Isstrøm reached maximum mean speeds, along the stacked dataset (averaged over period 2014–2017), of approximately 9 km yr$^{-1}$, 1.2 km yr$^{-1}$, 1.4 km yr$^{-1}$, 2.7 km yr$^{-1}$, respectively. The location of the velocity maxima varied between glaciers, as a result of their differing geometries. For JI and ZI, neither of which have a significant floating tongue, we find a progressive increase in ice velocity towards the calving front (Figures 2a and 2d). For PG, the maximum velocity is reached at the grounding line and remains steady along the ~46 km of ice tongue (Figure 2b). In contrast, although 79G also reaches its maximum velocity close to the grounding line, its speed then diminishes by ~ 50 % (Figure 2c) near the ice front location where the ice flow divides into two main portions before it reaches several islands and ice rises (Figure S1b). Furthermore, it is interesting to note that, despite being located in the same region, the adjacent glacier ZI flows ~60 % faster in comparison. JI, PG and ZI glaciers show velocity increases progressively downstream across the transverse profiles. The four glaciers, JI, PG, 79G and ZI respectively reduce their maximum velocity to half at distances of 12 km, 22 km, 18 km, and 12 km inland of their grounding lines, highlighting the importance of resolving glacier velocities within their near terminus regions.

Next, we used the Sentinel-1a/b and TerraSAR-X velocities to assess the seasonal and longer-term variations in Jakobshavn Isbræ ice velocity over the period 2009–2017. Our Sentinel-1a/b velocity estimates at JI resolve clear seasonal velocity fluctuations, superimposed upon longer term decadal scale variability, which continues observations made by previous satellite instruments [Joughin et al., 2012; Joughin et al., 2014]. At site J1 we find an average seasonal change in speed of 750 m yr$^{-1}$, or 14 % between 2014 and 2015 and a speedup persistence of 80-95 days, being twice longer than for the other three glaciers (Table 1). Inland, the amplitude of seasonal variability diminishes, to an average of 300 m yr$^{-1}$ (8 %) at J2. Our near-continuous, decadal-scale record clearly shows that the amplitude of the seasonal signal has evolved through time. At J1, for example, the average seasonal variability in ice speed was 400 m yr$^{-1}$ during 2009–2011, increasing by more than a factor of 3, to 1400 m yr$^{-1}$ between 2012 and 2013 and then diminishing to 750 m yr$^{-1}$ between 2015–2017.

Turning to the longer term evolution of JI (Figure 5; time series location shown in Figure 1), fitting a linear trend to the data suggests an annual acceleration since 2009 of ~218 m yr$^{-1}$ at Jif, diminishing inland to ~128 m yr$^{-2}$ at J1, and ~102 m yr$^{-2}$ at J2. Although this provides a simple characterisation of the longer-term evolution in ice speed, it is clear from our time series that computing a linear trend does not capture the full decadal scale variability in ice velocity. In particular, we note that much of the acceleration occurred between 2011 and 2013 (Figures 5b and 5c), and since then there has been a notable absence of multi-annual acceleration as earlier records suggest [Joughin et al., 2014]. Computing trends in ice velocity since 2012 near the glacier terminus (Jif), for example, shows a modest decline in speed of 321 m yr$^{-2}$ over the 5-year period (Figure 5b). The calving front position migration has been suggested as the trigger to the stresses regimes variations and consequently the main driver to the JI velocity fluctuations [Joughin et al., 2008a; 2008b; 2012; 2014; Bondzio et al., 2017]. After successive and gradually increased rate of the ice front retreat until 2012 (Figure 5a), the JI grounding line is now located on a higher bed location (Joughin et al., 2012; An et al., 2017). This may be acting to stabilise the grounding line, and in turn contribute to the glacier deceleration, although the main cause remains to be determined and further investigations is necessary. We used our observations of calving front position to assess the correlation between ice speed and calving front location, relative to their respective long term means (Figure 6). Based on the linear regression (Figure 6), our dataset indicates correlation coefficients ($R^2$) of 0.62 (2009–2011) and 0.79 (2012–2017), and velocity changes by 1100 and 1600 m yr$^{-1}$ per kilometre of calving front retreat, respectively.

At Petermann Glacier we extracted two velocity time series at P1, located ~45 km downstream of the grounding line and close to the calving front of the ice tongue; and P2, ~10 km upstream of the grounding line. These locations were chosen to examine any differences in velocity evolution over the grounded and floating portions of the glacier. Our P1 time series starts in early 2015 because it is not covered by the TerraSAR-X dataset (Figure 7a). We observe that, in general, ice at P1 flows ~400 m yr$^{-1}$ faster than P2. Fitting a linear trend to the longer P2 dataset indicates no significant trend in ice velocity since 2011, although the precision of this trend is hampered by the sparse data coverage during the early part of this period. Continued monitoring by Sentinel-1 will improve our confidence in resolving any decadal scale variability. The improvement in temporal sampling provided by Sentinel-1 at this site is clear (Figure 7a), allows us to resolve the seasonal cycle in velocity since 2015 and helps to delimit the duration of the speedup period. At P1, we detect a seasonal change in speed of ~ 300 m yr$^{-1}$, equivalent to a 25 % increase relative to its winter velocity (Table 1). Despite the high seasonal change, the relation between P1's annual mean and winter velocity is 0 %, likely due to the short speedup period (25 days - Table 1). This provides further evidence of a seasonal velocity cycle which has been observed at both Petermann and other glaciers in this region, and is understood to be predominantly controlled by changes in basal traction, induced by penetration of surface melt water to the bed [Nick et al., 2012; Moon et al., 2014; Moon et al., 2015]. This is further supported by our analysis of changes in calving front position (Figure S1a) which shows that, in contrast to JI, seasonal acceleration does not coincide with ice front retreat. Specifically, we found that during the summers of 2015 and 2016, the calving front of PG advanced ~1 km during the speedup (Figure S1a). These observations are consistent with previous modelling results, which did not find evidence of acceleration driven by large calving events in 2010 and 2012 [Nick et al., 2012; Münchon et al., 2014], suggesting that the ice shelf exerts low backstress on the glacier. More recently, we note that since September 2016 PG has developed a new crack near the ice front, which has continued to grow in length up to the present day.

At 79-G, we again extracted velocity time series over the ice shelf (F1, ~20 km downstream of the grounding line) and at the grounding line (F2). In contrast to PG and due to the steeper surface gradient upstream of the grounding line (Figure 2c), ice flow is slower on the floating tongue than at the grounding line location (Figure 7b). We observe a seasonal speed up of ~10 % at F2 during summer 2016 (Table 1), although evidence of the same acceleration on the ice shelf is not clear given the magnitude of the signal and the precision of our data. Fitting a linear trend to our data returns an annual change in velocity of 15 m yr$^{-2}$ since 2011, although assessing the significance of this result is difficult given the limited data sampling early in the period. Turning to Zachariæ Isstrøm, we extract time series at two locations slightly upstream of the grounding line in order to observe different temporal responses between them (Figure 7c). At this glacier, no observations are available within the TSX dataset and so our time series is limited to the period December 2015 to January 2017. Nonetheless, like its neighbour ZI, we again find evidence of a summer speed up during 2016, equating to around 400 m yr$^{-1}$, or 18 % (Table 1). Given the short period of observations we do not attempt to derive a longer-term trend in ice velocity at this site.

We compared our estimates to the results of previous studies to assess the level of stability relative to past work. At Petermann, we have observed increases in ice velocity of ~10 % at P1 and ~8 % at P2 between the 2015/2016 and 2016/2017 winters, matching in percentage with the observations made by Münchon et al. [2016] between 2013/14 and 2015/16. Furthermore, the Sentinel-1a/b dataset indicates a multi-annual acceleration of ~32 m/yr$^2$ between 2015-2017 at P1, which is similar to the ~30 m/yr$^2$ reported by Nick et al. [2012] based upon observational measurements over a longer period, from 2006 to 2010. The same authors also show seasonal variations of ~20–25 % over the same location, similar to the ~22 % shown by the Sentinel-1 dataset. At 79-G, Mouginot et al. [2015] showed a speedup of 8 % from 1976 to 2014 with the main changes occurring after 2006, similar to our estimates which also suggest a slight multi-year trend of ~16 m yr$^{-2}$ (~8 %) for F2 between 2015 and 2017. Zachariæ Isstrøm shows seasonal variation up to 15 % between 2015 and 2017 in the Sentinel-1 dataset, agreeing with seasonal variation up to 20 % estimated by Mouginot et al. [2017] using Landsat-8 optical images during 2014–2016. Overall, our Sentinel 1 results shows a close agreement with previous studies using different techniques and demonstrated to be a powerful tool for monitoring the cryosphere.

# 6        Conclusions

We have presented a new, high temporal resolution record of ice velocity evolution for four important, and with high discharge, marine based glaciers in Greenland, updated to the present day (October 2014 to February 2017). Using SAR data acquired by the Sentinel-1a/b constellation, with its 250 km wide swath and frequent revisit time, we have produced 187 velocity maps, which, in combination with 479 maps from the TerraSAR-X satellite, provide detailed spatial and temporal coverage of these key sites. Importantly, the systematic acquisition cycle of Sentinel-1a/b, which now provides averaged measurements of all of these sites every 6 days allows for detailed monitoring of both seasonal and multi-annual velocity fluctuations, and allow us to demonstrate the speedup persistence in a higher resolution. The short revisit time of 6 days, made possible since the launch of Sentinel-1b in April 2016, particularly benefits the retrieval of velocity signals across fast flowing regions close to the ice front, due to a reduction in the decorrelation occurring between image pairs. Using this new dataset, we confirm evidence of intra-annual variations in ice velocity and clear seasonal cycles occurring over the past few years at JI, PG, 79G and ZI. Of the sites studied here, JI exhibits the largest velocity variations, as demonstrated in other studies, which we show are strongly correlated with the evolution of the position of its calving front. Notably, however, in the last 5 years the longer-term ice speed has started to decrease (321 m yr$^{-2}$). This study demonstrates the utility of a new era of operational SAR imaging satellites for building systematic records of ice sheet outlet glacier velocity and its good agreement with TerraSAR-X products, which indicates Sentinel-1 can confidently extend the times series that began with other sensors. Looking to the future, these datasets are key for the timely identification of emerging signals of dynamic imbalance, and for understanding the processes driving ice velocity change.

*Competing interests*. The authors declare that they have no conflicts of interest.

*Acknowledgements*: This work was led by the NERC Centre for Polar Observation and Modelling, supported by the Natural Environment Research Council (cpom300001), with the support of a grant (4000107503/13/I-BG), and the European Space Agency. AEH is funded from the European Space Agency's support to Science Element program, and an independent research fellowship (4000112797/15/I-SBo). A.L. was supported by CAPES-Brazil PhD scholarship. The NASA MEaSUREs program (NASA grantsNNX08AL98A and NNX13AI21A) supported I.J.'s contribution. All data is freely available for download at http://www.cpom.ucl.ac.uk/csopr/iv/. We thank the editor Andreas Vieli, and the two reviewers, Rachel Carr and an anonymous referee, for their comments, which helped to improve the manuscript.

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

## 7    Figures and Captions

- Figure 1. Time-averaged ice velocity magnitude maps for the period Oct/2014–Feb/2017 (a) Jakobshavn Isbræ (JI; 69°N, 50°W), (b) Petermann Glacier (PG; 81°N, 62°W), (c) Nioghalvfjerdsfjorden (79G; 79°N, 20°W) and Zachariæ Isstrøm (ZI; 78°N, 20°W) glaciers, derived from Sentinel-1 SAR images. Velocities are shown on a logarithm scale and overlaid on a SAR backscatter intensity image and thin grey lines represent elevation. The along-flow profiles are indicated by solid green lines scaled in kilometres, the solid black lines show the across-flow transects, the red triangles represent the locations at which velocity time series are extracted and the thick solid and dashed black lines represent the ice front locations (IF) and the grounding lines (GL), respectively. The inset figures show the location of each glacier.

- Figure 2. Average velocities (2014–2017) extracted from along- and across-flow profiles of Jakobshavn Isbræ, Petermann Glacier, Nioghalvfjerdsfjorden and Zachariæ Isstrøm. Figures a–d present along-flow profiles of ice velocity (solid black lines), surface elevation from the GIMP DEM [Howat et al., 2014; dashed blue lines] and bed elevation from the IceBridge BedMachine Greenland V2 product [Morlighem et al., 2015; dashed yellow lines]. The location of each profile is shown in Figure 1 (green lines). The grey shaded area represents the floating regions, and the light grey dashed line the ice front positions. The blue, black and red markers represent the locations of the across-flow profiles. Figures e–h show the across-flow velocity profiles (solid white lines in Fig.1), centred on the main profile (solid green line).

- Figure 3. Time-averaged (2014–2017) uncertainty in ice velocity at each site expressed in percentage, based on the signal to noise ratio (SNR) for (a) JI, (b) PG, and (c) 79G and ZI.

- Figure 4. Comparison between co-located and contemporaneous Sentinel 1-a/b (6 to 12 days average) and TerraSAR-X (11 days average) Jakobshavn Isbræ velocity measurements at Jif, J1 and J2 locations (blue, black and red dots respectively), together with root mean square (rms) and correlation coefficients ($R^2$).

- Figure 5. Temporal evolution of Jakobshavn Isbræ (a) ice front position extracted from Joughin et al. [2014], ESA Greenland Ice Sheet Climate Change Initiative (CCI) project [2017], and Sentinel-1a/b SAR images represented in blue, black and magenta dots respectively, where higher values correspond to ice front retreat. Changes in ice velocity through timeis also shown (b, c), extracted at the locations indicated in Figure 1. The velocity data derived from TerraSAR-X (11 days - Joughin et al., 2016) are shown as grey squares, and the data from Sentinel 1-a/b (6 to 12 days) as coloured triangles.

- Figure 6. Comparison between Jakobshavn Isbræ ice velocity and calving front position anomalies at the Jif site, 0.8 km upstream of the calving front, between 2009 and early 2017. Positive values correspond to ice front retreat and speed up respectively. The red and black lines represent the linear regression through the 2009-2011 and 2012-2017 periods, respectively, together with the correlation coefficients ($R^2$).

- Figure 7. Temporal evolution of ice velocity at the locations indicated in Figure 1 over (a) Petermann Glacier, (b) Nioghalvfjerdsfjorden and (c) Zachariæ Isstrøm. The data derived from TerraSAR-X (11 days - Joughin et al., 2016) and Sentinel 1-a/b (6 to 12 days) are represented as grey squares and coloured triangles, respectively.

- Table 1: Speedup Persistence and seasonal percentage increase in speed relative to winter and annual background for each glacier for the Sentinel 1 dataset. Speedup persistence has an uncertainty of ± 12 days due to the image acquisition interval of Sentinel 1a.

Figure 1

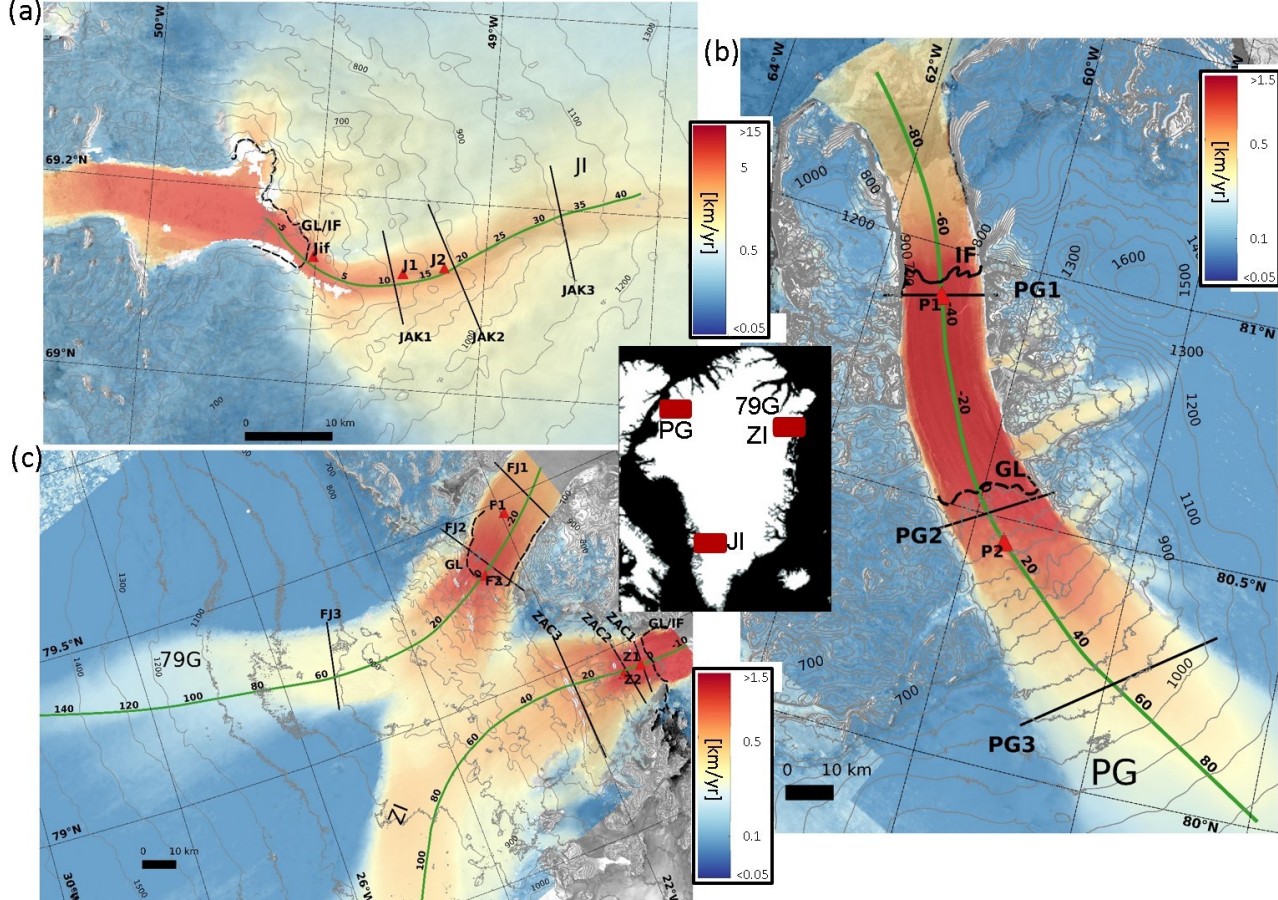

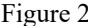

Figure 2

(a) **Jakobshavn Isbræ**

(b) **Petermann**

(c) **Nioghalvfjerdsfjorden**

(d) **Zachariæ Isstrøm**

(e)

(f)

(g)

(h)

Figure 3

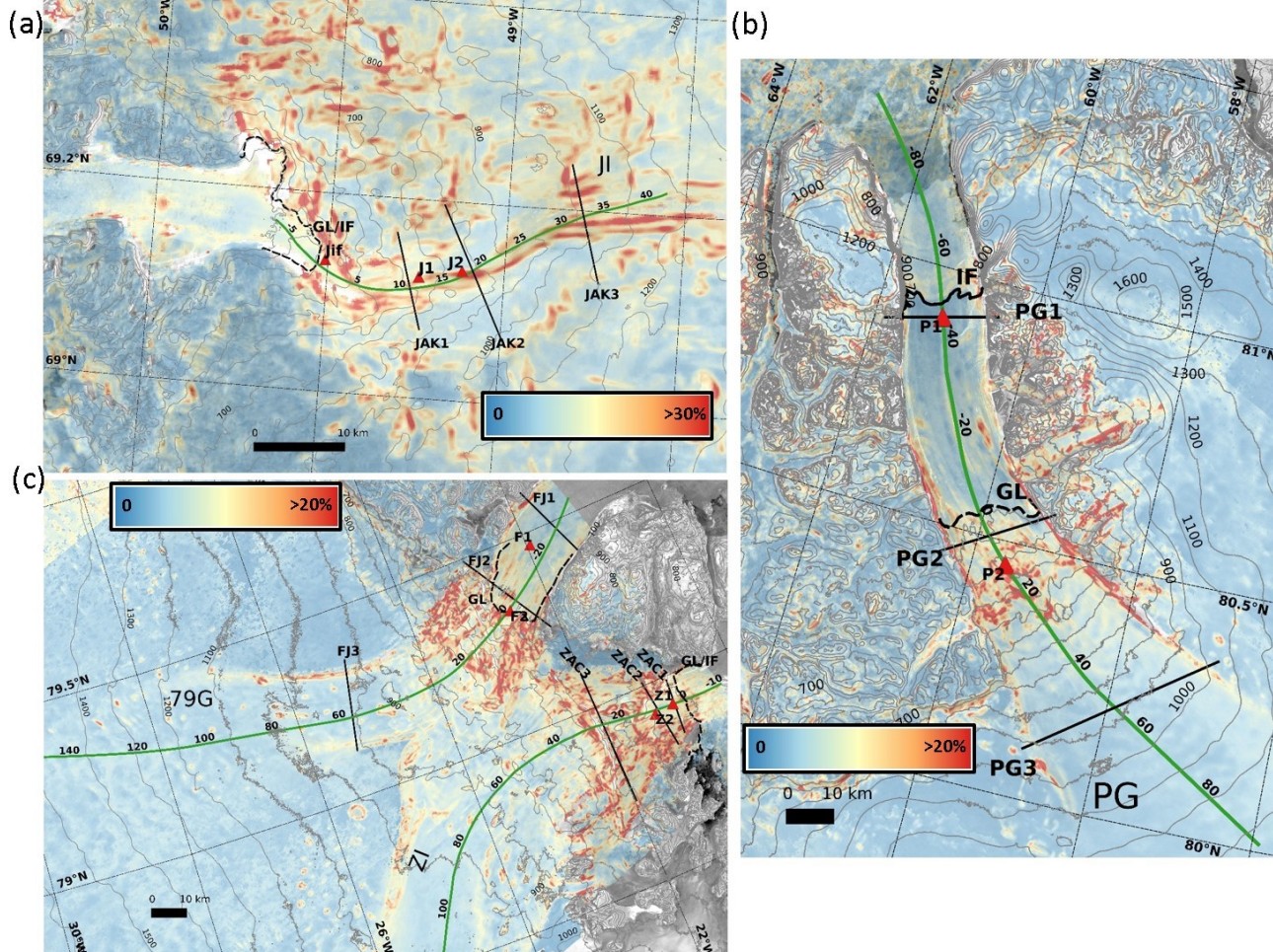

Figure 4

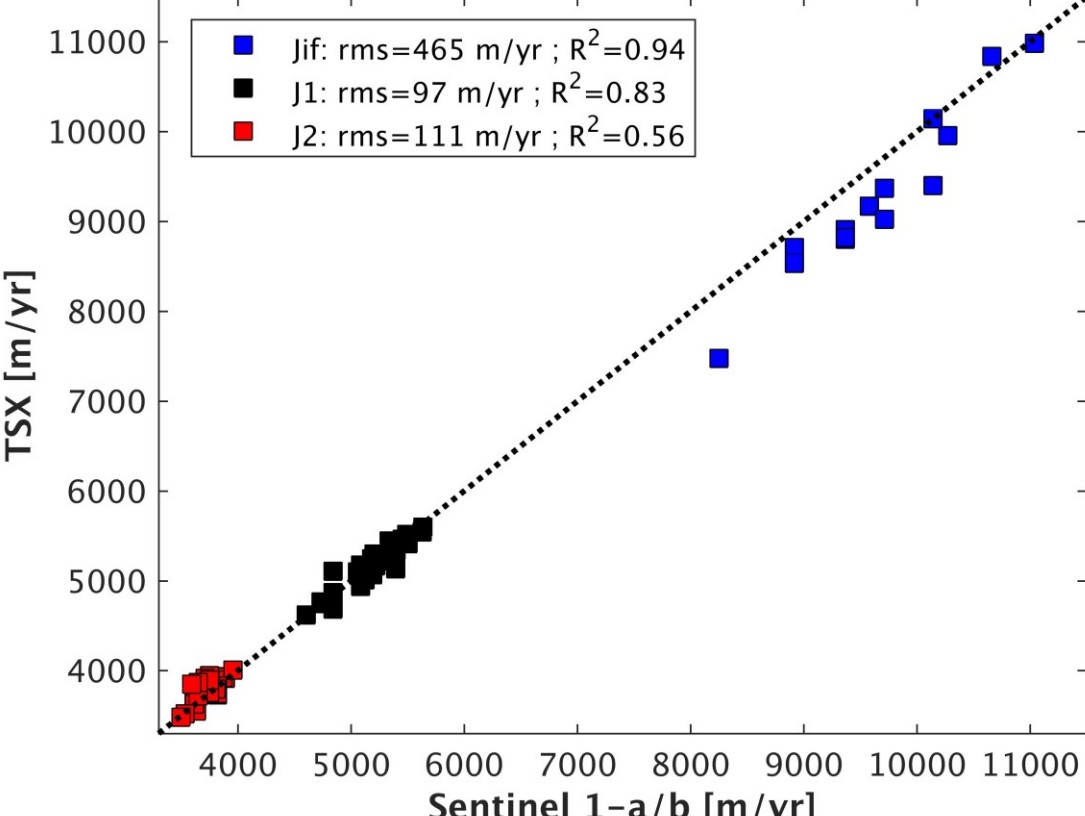

Figure 5

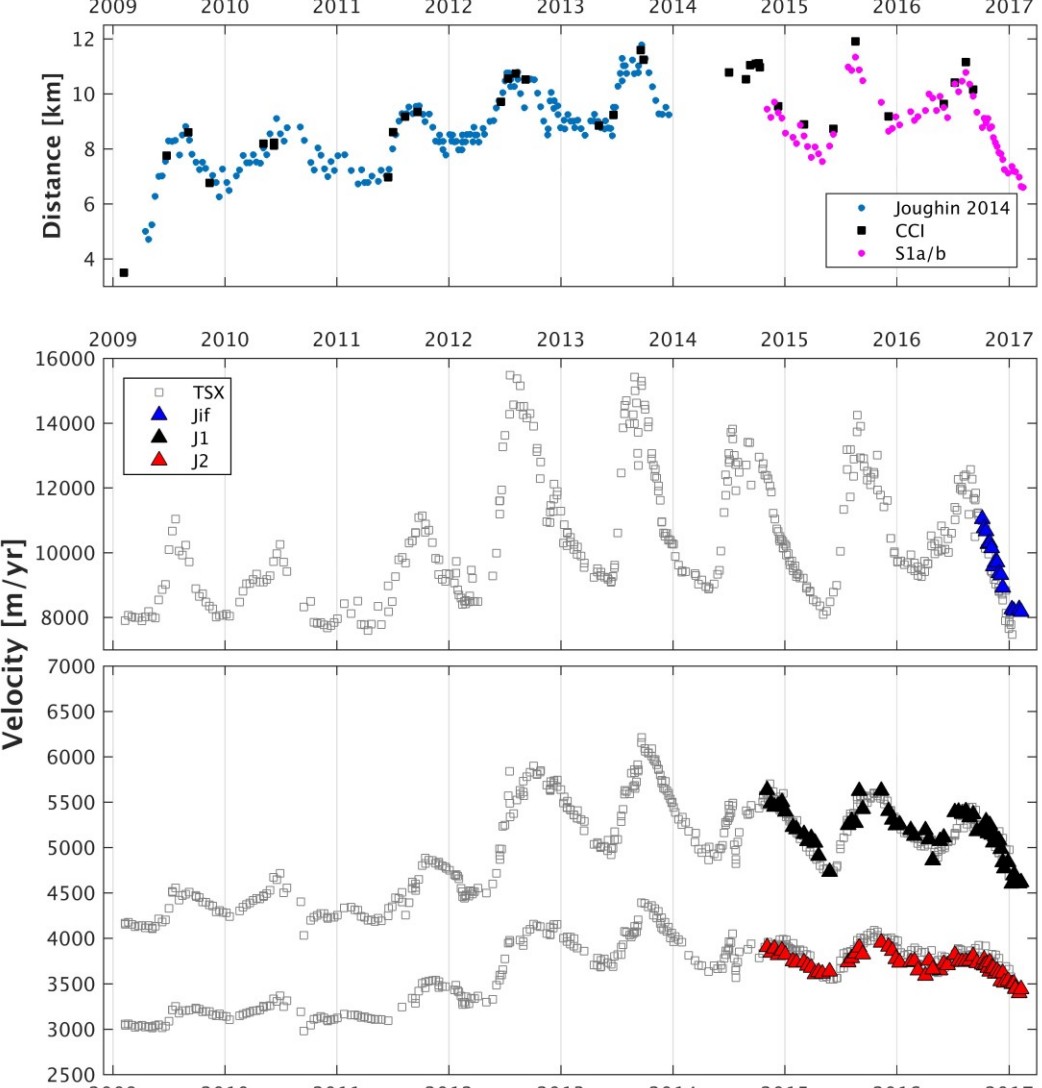

Figure 6

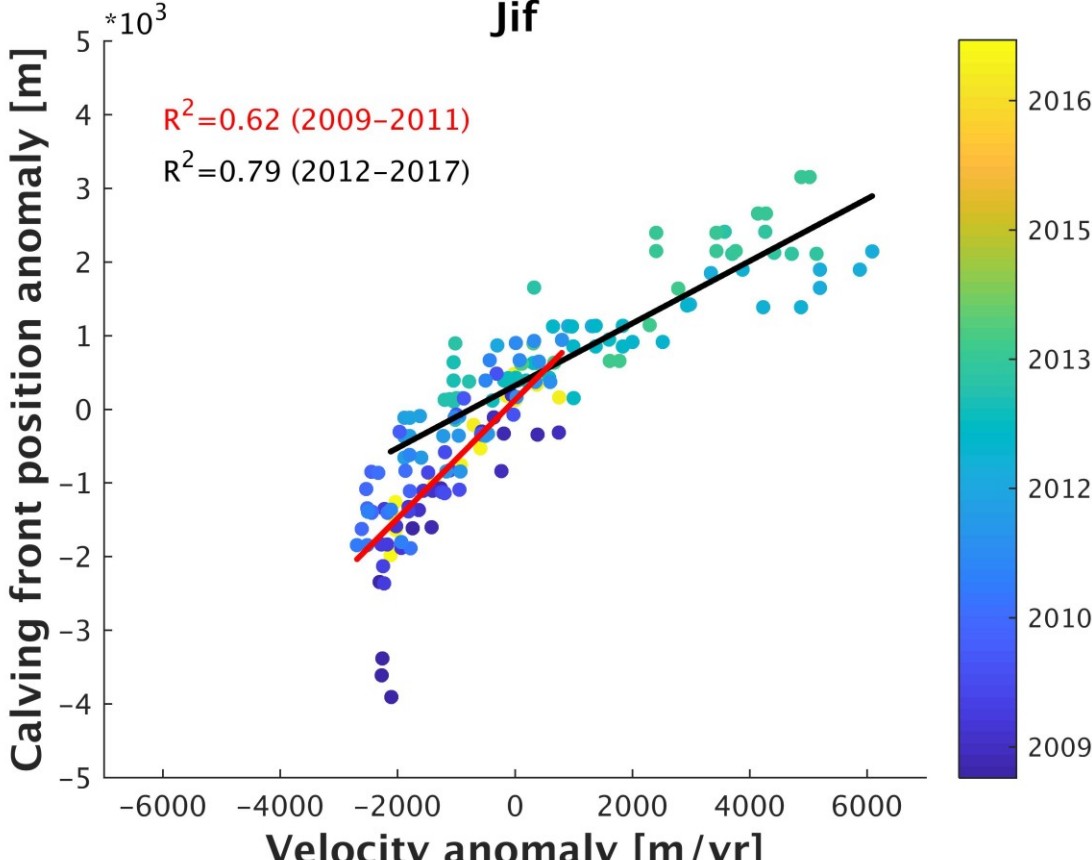

Figure 7

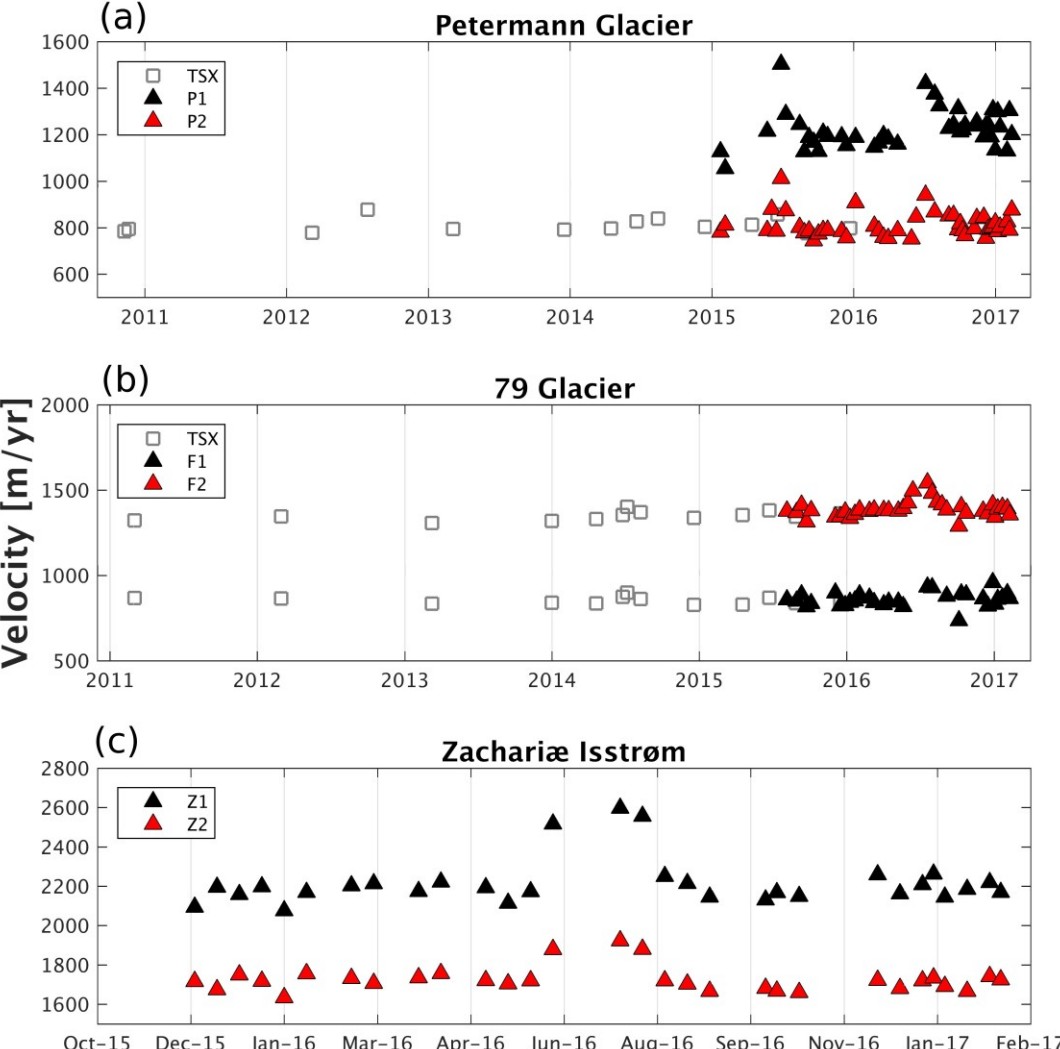

Table 1

| | Speedup Persistence | Summer speedup (%) | $V_{annual}/V_{winter}$(%) |
|---|---|---|---|
| **JI (J1)** | 95 days (2015)<br>80 days (2016) | 14%<br>9% | 6%<br>4% |
| **PG (P1)** | 25 days (2015)<br>55 days (2016) | 25%<br>17% | 0%<br>6% |
| **79G (F2)** | 45 days (2016) | 10% | 1% |
| **ZI (Z1)** | 45 days (2016) | 18% | 3% |