# Peer review of "Ice velocity of Jakobshavn Isbræ, Petermann Glacier, Nioghalvfjerdsfjorden and Zachariæ Isstrøm, 2015-2017, from Sentinel 1-a/b SAR imagery"

_The Cryosphere, 2017_

## Referee Comment (RC1) · Anonymous Referee #1 · 20 Feb 2018

Review of "Ice velocity of Jakobshavn Isbræ, Petermann Glacier, Nioghalvfjerdsfjorden and Zachariæ Isstrøm, 2015-2017, from Sentinel 1-a/b SAR imagery" by Lemos et al.

MS : tc-2017-251

Summary: The authors present velocity data from Sentinel 1-a/b from four major Greenland Ice Sheet outlet glaciers. The data provide higher temporal resolution than existing datasets and extend the existing temporal record by several years. The authors document multi-annual and inttra-annnual speed changes.

[Figure]

General comments

1) The paper could be significantly improved by expanding on the implications of observations for better understanding ice dynamics. Right now, to me it reads as a lot of results without much discussion of significance or implications. For this work to be published in The Cryosphere, it seems it should increase our understanding of glacier mechanics/dynamics, not just describing what we see. For example, why has Jakobshavn Isbrae (JI) begun slowing? Why did the amplitude of seasonal velocity change increase on JI at the same time?

2) It seems to me that you could better exploit the novelty of this high temporal resolution dataset to investigate processes at "the timescales over which glacier dynamics evolve". Resolving multi-annual trends in velocity doesn't require 6 or 12 day repeat times. But investigating seasonal dynamics hugely benefits from this increased temporal resolution. I believe your paper could be strengthened (in its ability to demonstrate capability of new generation radar satellites) by digging in deeper to these processes that are more difficult to resolve with existing datasets.

3) The introduction could use more substance by referencing relevant existing work. See "specific comments" for some suggested references.

4) I wonder if your method for characterizing uncertainty may overestimate your error. It seems that by using SNR (where SNR = (mean velocity)/(standard deviation of velocity) within your window, correct?) would be higher where velocity is spatially variable. In this case, SNR would be high, but not because of bad data – because of physically meaningful velocity variation. It seems like the margins/shear zones typically have high uncertainty – could this just be because there are very different (physically meaningful) velocities across these regions?

For these reasons, and the specific comments listed below, I recommend the manuscript requires Major Revision before publication.

[Figure]

Specific comments

Page 1 L28: Your definition of mass balance appears to ignore negative terms in SMB (i.e., only mentions mass input)

L39: See below for relevant papers for this idea that should be cited – e.g.,

Felikson, D., Bartholomaus, T. C., Catania, G. A., Korsgaard, N. J., Kjær, K. H., Morlighem, M., . . . Nash, J. D. (2017). Inland thinning on the Greenland ice sheet controlled by outlet glacier geometry. Nature Geoscience, 10, 366–369. https://doi.org/10.1038/ngeo2934

Durkin, W. J., Bartholomaus, T. C., Willis, M. J., Pritchard, M. E. (2017). Dynamic changes at Yahtse Glacier, the most rapidly advancing tidewater glacier in Alaska. Frontiers in Earth Science, 5(March), 1–13. https://doi.org/10.3389/feart.2017.00021

Page 2 L7: Should cite modern landsat efforts mapping glacier velocity velocity change at large scale – e.g.,

Armstrong, W. H., Anderson, R. S., Fahnestock, M. A. (2017). Spatial patterns of summer speedup on south central Alaska glaciers. Geophysical Research Letters, 44. https://doi.org/10.1002/2017GL074370

Burgess, E. W., Forster, R. R., Larsen, C. F. (2013a). Flow velocities of Alaskan glaciers. Nature Communications, 4, 2146. https://doi.org/ 10.1038/ncomms3146

Dehecq, A., Gourmelen, N., Trouve, E. (2015). Deriving large-scale glacier velocities from a complete satellite archive: Application to the Pamir-Karakoram-Himalaya. Remote Sensing of Environment, 162, 55–66. https://doi.org/10.1016/j.rse.2015.01.031

Fahnestock, M., Scambos, T., Moon, T., Gardner, A., Haran, T., Klinger, M. (2016). Rapid large-area mapping of ice flow using Landsat 8. Remote Sensing of Environment, 185, 84–94. https://doi.org/10.1016/j.rse.2015.11.023

Jeong, S., Howat, I. M. (2015). Performance of Landsat 8 Operational Land Imager for mapping ice sheet velocity. Remote Sensing of Environment, 170(8), 90–101. https://doi.org/10.1016/j.rse.2015.08.023

L27-28: Many authors have shown that high melt years actually correspond with smaller net annual displacements – e.g.,

Burgess, E. W., Larsen, C. F., Forster, R. R. (2013b). Summer melt regulates winter glacier flow speeds throughout Alaska. Geophysical Research Letters, 40, 6160–6164. https://doi.org/10.1002/2013GL058228

Tedstone AJ and 6 others (2013) Greenland ice sheet motion insensitive to exceptional meltwater forcing. Proc.Natl. Acad. Sci.U. S. A., 110(49), 19719–19724 (doi: 10.1073/pnas.1315843110)

Van De Wal, R. S. W., Smeets, C. J. P. P., Boot, W., Stoffelen, M., Van Kampen, R., Doyle, S., . . . Hubbard, A. (2015). Self-regulation of ice flow varies across the ablation area in south-west Greenland. Cryosphere, 9(2), 603–611. https://doi.org/10.5194/tc-9-603-2015

Page 3 L35: I believe you are talking about 5 cm position uncertainty in the satellite. What is your uncertainty in the ground coordinates of each pixel?

L36-37: I am not a SAR person – do you have to model or account for atmospheric delays? Seems like you would have to do that before isolating glacier displacement.

L41-42: What year was the data used for the GIMP DEM collected? How would error in the GIMP DEM (or elevation change between its acquisition and your time series) affect your velocity estimates?

Page 4 L4-6: What labelling algorithm do you use? This bit is unclear and would be difficult to repeat.

L10-13: It seems like this approach would lump actual physical spatial velocity variability with spurious non-physical velocities. Could this be why your glacier margins/shear

zones on Figure 3 are always high uncertainty? There actually is high spatial variability in velocity there, so SNR would be low, but not because of bad data. It seems like this method may overestimate error in some regions.

L17: What is your "airborne estimate of elevation change"? What constitutes an "extreme case" (i.e., what is the magnitude "large" of surface lowering? Needs more detail.

L15-24: Are these errors systematic or random? Seems like tidal forcing may be random (sometimes surface is higher, other times surface is lower than you think) but thinning would be systematic (always lower than you think).

L26-27: It seems that using one SAR estimate of velocity to check another SAR estimate of velocity is vulnerable to errors that would effect both in the same way. Do you have any known velocities from GPS or optical image correlation to compare against?

L26-27: Should also specify that TSX is based off phase change (InSAR) and you are using feature tracking of SAR imagery (amplitude based) if that is indeed the case. This provides more evidence that using it is a robust check on your data because it is a different method.

L 7-9: How does this "highlight the importance of resolving glacier velocities within their near terminus regions"? And resolve for what purposes?

L25: What station is this 5 year speed change calculated at? Jif? Looks like JI1/2 are relatively stable over this time. What would magnitude be if calculated there?

L1 - This is interesting that you find different relationship between speedup and retreat at different glaciers. But seems similar to findings by Moon et al., that find some glaciers terminus-forced, others see more land-terminating style of meltwater-forced. If dL = u$_i$ce −

$-u_calve, then would expect lengthening if calving can't keep up with faster ice flow in summer.$

L20-24 – How could Munchon et al. [2016] have used data from 2016/2017? And seems like comparing apples to oranges to compare speedup calculated over different time spans. That seems to tell more about glacier dynamical change than measurement accuracy.

Figure 1 – difficult to see lines. Please thicken all lines, but especially green. I also think the overlay makes the image harder to interpret. Maybe just overlay on a grayscale hillshdaded DEM that would provide topographic data but not confuse velocity information.

Figure 2 – This figure would be clearer to me if you labeled the lines in at least panel a

Figure 5 – Could you show earlier data (from a different source) to put this plot in context of longer-term JI velocity evolution?

Figure 6 – How are portions of data for fitting red and black lines decided? Don't think either is the 2009-2011 or 2012-2017 fit lines mentioned in the text?

Technical corrections – typographical errors

L33: language is a little sloppy/unclear – marine terminating glaciers still have SMB – should specify this 30

L34-35: "erosion of their termini" → replace with "terminus retreat" or "submarine melting"; I can't tell what you mean and "erosion" could be confused for subglacial bedrock lowering

L38: "high frequency variability" → "high spatial variability" if this is what you mean

L8: "Polar Regions" → "polar regions"

L26: "over the past few years" implies the JI speedup is ongoing (which you show it is

not)

L33-35: Sentence starting with "Therefore. . ." is confusing. You are saying -10 km2/a is not a big area change? Could just reword this sentence to be positive (e.g., "glacier area has remained relatively constant") instead of negative (e.g., "glacier area has not changed an unusually large amount")

L1: migration of what? Would call this "terminus retreat rate" if that is indeed what it is.

L8: Sentence starting with "Although located. . ." could be reworded to use fewer commas.

---

## Referee Comment (RC2) · J.R. Carr (Referee) · 12 Mar 2018

This paper uses new, high-resolution satellite imagery to assess velocity variations on four large Greenlandic outlet glaciers. Overall, it is very well written, clear and topical. It makes a useful contribution to the field and presents interesting results. It also nicely illustrates the usefulness of these datasets. It is concise, but addresses the questions it sets out to answer. I have a few minor comments below, which are noting places where things could be clarified or expressed more clearly. Overall, however, I think it is a really good paper in its present form. Nice to see such a well-put together and

concise paper.

Line by line comments Page 1 L13: Indicate temporal resolution in brackets. L19: Give date. L30: Sentence is a bit hard to follow. Consider splitting. L34: Specify the time period as 'last years' will date with the paper. L37: There are other references that are relevant here, e.g. Jensen et al., 2016, Carr et al., 2017 (J Glac). There are a couple of places in the intro where only one or two refs are given, but there are clearly more. Please add a selection of relevant ones and add 'e.g.' to indicate awareness of the others you don't list. Also need a couple of references for the statement about glacier specific and climatic controls.

Page 2 L1: Good justification for the study. L3: ...ice sheet dynamics AND ice discharge, and for assessing.... L14: Why these glaciers. I'd add a sentence of two. This is sort of given in the next sentence, but it feels like it needs a clear justification at this point. L33: Seems odd to switch to area change here, after discussing retreat. I know this paper does look at area change, so I'd work on improving the flow of the argument here.

Page 3 L21: Is this the maximum data range? Worth stating for clarity. L26: Might be useful to have graph showing image availability for each glaciers over time. E.g. you could have a bar graph, with number of images on the Y axis, then time on the x Axis, at monthly intervals. You'd then have a different coloured bar for each of the four glaciers. I'm suggesting this so that the reader can get a better handle on how these 187 velocity maps are distributed over the glaciers and over time. L40: Why was this value used?

Page 4 L2: Why was this spacing used? Was it because of the GIMP resolution? L14: Please state what these higher errors were in a separate sentence. L43: What period were these means taken over? Or do you mean the maximum value of the means across the whole glacier? Generally this is done well, but make sure you give the time period / spatial extent of your averages, as it's sometimes hard to follow which average

/period you are discussing. L44: Do you mean velocity MAXIMA (not magnitude)?

Page 5 On this page, there are several points were it would be useful to refer back to the relevant figures, e.g. L2, L24, L26. Please update throughout, as it helps guide the reader quickly to the relevant figures. L2: I'm not clear what '46 km extension' refers to. Please revise this description so it's clear. L7: I find this hard to follow. Please re-phrase. L12: ...scale variability, which continues..... L14: Helpful to repeat the time period these values relate to. L34: starts IN early 2015.

Page 6 L10: Please state whether or not this relationship is statistically significant, then have the explanation of why it is not significant (i.e. lack of data). L27: I think it would help to add a summary sentence or two. I'd definitely add one to sum up the key message of this paragraph (i.e. your data agree pretty well), and maybe add another, more general summary sentence to reflect on the usefulness of the data for this purpose. At the moment, it feels like the paper ends abruptly, even if you do say this in the conclusion. L31: Be specific about what you mean by 'important', I.e. high flow, large discharge. L32: I'd just give the date, as 'the present day' dates.

Figures Figure 1 & 3: The labels on these maps need to be much bigger, especially the locations of the extracted velocities. I find it really difficult to see these, but they're important for the context of the paper. I also think the three Greenland overviews are too small and don't work. Instead, please add one Greenland overview, with the sites marked, but which is a reasonable size. It's a shame to have nice figures like these when the reader can't read them properly. Figure 2: As with the other figures, this needs to be bigger, especially the text, as I can barely read it, especially the axes labels. Figure 3: The grounding line needs to be much more obvious, as do the lines for the termini. I really struggle to see them. Same for figure 1 Particularly for A), it would be useful to have a land mask to orientate people. Figure 4: make the line stronger ad markers larger. Add the p-value and R2 for this regression. Figure 5: The text is much easier to read on this, but some of the points are hard to see, e.g. the green dots in A. I know it's difficult given the data density, but think how you can back

the points easier to see throughout this figure (e.g. through increasing size or changing the colours). Figure 6: Add p-values for the regression lines. Figure 7: Might be clearer with slightly large points

Overall, the figures are really good and illustrate the points well, but you have to make sure they are readable, otherwise all of the hard work is wasted!

In summary, this is a really nice, concise paper, and I really enjoyed reading it. Thanks!
* * *

---

## Author Comment (AC1) · 12 May 2018

12th May 2018

Centre for Polar Observation and Modelling

School of Earth and Environment

University of Leeds

5   LS2-9JT, Leeds - UK

Dear Prof. Vieli,

**"Ice velocity of Jakobshavn Isbræ, Petermann Glacier, Nioghalvfjerdsfjorden and Zachariæ Isstrøm, 2015-2017, from Sentinel 1-a/b SAR imagery"**

Thank you for considering the above manuscript for publication in The Cryosphere. We have submitted a
15   revised manuscript that addresses each of the issues raised by the reviewers, and accompanying documents
that describe these changes in detail.

We have given particular attention to the major issues raised by Reviewer 1, and full details of the changes
and our response to these points are given below. In summary, we have (1) added more glaciological
interpretation of the data focused on the capability offered by the high temporal sampling rate, namely to
20   resolve the magnitude and duration of summer speed up at each glacier, (2) added further discussion relating
to the glaciological interpretation of the data, (3) revised and expanded the introduction, including adding
relevant references, and (4) clarified our method for estimating uncertainty.

We are grateful for the comments provided by yourself and the reviewers, as they have helped to substantially
improve the manuscript, and we hope that the changes are to your satisfaction. We look forward to your reply.

25   Yours faithfully

Adriano Lemos

(corresponding author)

**INTERACTIVE COMMENTS**

The responses (A) to the referees' comments are shown in blue below.

**ANONYMOUS REFEREE #1**

We thank the anonymous referee for the comments and suggestions, which have significantly improved the manuscript.

**GENERAL COMMENTS:**

R#1: 1: The paper could be significantly improved by expanding on the implications of observations for better understanding ice dynamics. Right now, to me it reads as a lot of results without much discussion of significance or implications. For this work to be published in The Cryosphere, it seems it should increase our understanding of glacier mechanics/dynamics, not just describing what we see. For example, why has Jakobshavn Isbrae (JI) begun slowing? Why did the amplitude of seasonal velocity change increase on JI at the same time?

A: We have modified the abstract and discussions to make clear the scope of the paper, and to add some preliminary discussion relating to the origin of the JI slowdown. However, we emphasise that the purpose of this study was to develop and report new observations of glaciological change, rather than to provide a detailed process-orientated investigation. The former in itself, we believe, is a significant body of work, which reports new and important findings that will be of interest to a broad spectrum of the cryospheric community, and is therefore relevant for publication in The Cryosphere. The slowdown of JI, for example, is an important new finding and we have extended our discussion of the possible cause within the revised manuscript (P5L35-40). However, a full process-based investigation of the associated forcing mechanisms is, we believe, well beyond the scope of this current study.

R#1: 2: It seems to me that you could better exploit the novelty of this high temporal resolution dataset to investigate processes at "the timescales over which glacier dynamics evolve". Resolving multi-annual trends in velocity doesn't require 6 or 12 day repeat times. But investigating seasonal dynamics hugely benefits from this increased temporal resolution. I believe your paper could be strengthened (in its ability to demonstrate capability of new generation radar satellites) by digging in deeper to these processes that are more difficult to resolve with existing datasets.

A: Thanks for the suggestion. We have added new analysis of the 6 day dataset to quantify the relative magnitude and duration of the seasonal velocity acceleration for all glaciers. We now add these results (table 1) as one of the main conclusions of the paper, and report these findings in the abstract and the discussion section.

R#1: 3: The introduction could use more substance by referencing relevant existing work. See "specific comments" for some suggested references.

A: Thank you for the comments and the extras references suggested. We have rewritten unclear points and added more references. Further details are provided in the responses to specific comments below.

R#1: 4: I wonder if your method for characterizing uncertainty may overestimate your error. It seems that by using SNR (where SNR =(mean velocity)/(standard deviation of velocity) within your window, correct?) would be higher where velocity is spatially variable. In this case, SNR would be high, but not because of bad data – because of physically meaningful velocity variation. It seems like the margins/shear zones typically have high uncertainty – could this just be because there are very different (physically meaningful) velocities across these regions?

A: The signal to noise ratio (SNR) used in this work is a relation between the cross-correlation function peak (Cp) and the average correlation level (CL) on the tracking window used to estimate the velocity (SNR=Cp/CL). We clarify this point on lines P4L16-17. The shear margins have high uncertainties because they are challenging areas to track velocity

due to the non-uniform flow, lack of stable features and geometry distortion even after the terrain correction (P4L19-21).

**SPECIFIC COMENTS:**

R#1: P1L28: Your definition of mass balance appears to ignore negative terms in SMB (i.e., only mentions mass input)

A: Done (P1L27-28).

R#1: P1L39: See below for relevant papers for this idea that should be cited – e.g.,

Felikson, D., Bartholomaus, T. C., Catania, G. A., Korsgaard, N. J., Kjær, K. H., Morlighem, M., . . . Nash, J. D. (2017). Inland thinning on the Greenland ice sheet controlled by outlet glacier geometry. Nature Geoscience, 10, 366–369. https://doi.org/10.1038/ngeo2934

Durkin, W. J., Bartholomaus, T. C., Willis, M. J., Pritchard, M. E. (2017). Dynamic changes at Yahtse Glacier, the most rapidly advancing tidewater glacier in Alaska. Frontiers in Earth Science, 5(March), 1–13. https://doi.org/10.3389/feart.2017.00021

A: We added additional references as showed in P1L39-40.

R#1: P2L7: Should cite modern Landsat efforts mapping glacier velocity changes at large scale – e.g.,

Armstrong, W. H., Anderson, R. S., Fahnestock, M. A. (2017). Spatial patterns of summer speedup on south central Alaska glaciers. Geophysical Research Letters, 44. https://doi.org/10.1002/2017GL074370

Burgess, E. W., Forster, R. R., Larsen, C. F. (2013a). Flow velocities of Alaskan glaciers. Nature Communications, 4, 2146. https://doi.org/10.1038/ncomms3146

Dehecq, A., Gourmelen, N., Trouve, E. (2015). Deriving large-scale glacier velocities from a complete satellite archive: Application to the Pamir-Karakoram-Himalaya. Remote Sensing of Environment, 162, 55–66. https://doi.org/10.1016/j.rse.2015.01.031

Fahnestock, M., Scambos, T., Moon, T., Gardner, A., Haran, T., Klinger, M. (2016). Rapid large-area mapping of ice flow using Landsat 8. Remote Sensing of Environment, 185, 84–94. https://doi.org/10.1016/j.rse.2015.11.023

Jeong, S., Howat, I. M. (2015). Performance of Landsat 8 Operational Land Imager for mapping ice sheet velocity. Remote Sensing of Environment, 170(8), 90–101. https://doi.org/10.1016/j.rse.2015.08.023

A: Done (P2L7-8).

R#1: P2L27-28: Many authors have shown that high melt years actually correspond with smaller net annual displacements – e.g.,

Burgess, E. W., Larsen, C. F., Forster, R. R. (2013b). Summer melt regulates winter glacier flow speeds throughout Alaska. Geophysical Research Letters, 40, 6160–6164. https://doi.org/10.1002/2013GL058228

Tedstone AJ and 6 others (2013) Greenland ice sheet motion insensitive to exceptional meltwater forcing. Proc.Natl. Acad. Sci.U. S. A., 110(49), 19719–19724 (doi: 10.1073/pnas.1315843110)

Van De Wal, R. S. W., Smeets, C. J. P. P., Boot, W., Stoffelen, M., Van Kampen, R., Doyle, S., Hubbard, A. (2015). Self-regulation of ice flow varies across the ablation area in south-west Greenland. Cryosphere, 9(2), 603–611. https://doi.org/10.5194/tc-9-603-2015

A: We rephrase the sentence and added one of the references (P2L29-31).

R#1: P3L35: I believe you are talking about 5 cm position uncertainty in the satellite. What is your uncertainty in the ground coordinates of each pixel?

A: We modified the text, making clear that the precise orbit establishes a 5 cm 3D 1-sigma co-registration accuracy requirement.

R#1: P3L36-37: I am not a SAR person – do you have to model or account for atmospheric delays? Seems like you would have to do that before isolating glacier displacement.

A: Our uncertainty estimates account other error's resources including the atmospheric delay due to ionospheric disturbances and/or tropospheric water vapour.

R#1: P3L41-42: What year was the data used for the GIMP DEM collected? How would error in the GIMP DEM (or elevation change between its acquisition and your time series) affect your velocity estimates?

A: Please find more information about GIMP digital elevation model in Howat et al. (2014). We discuss the elevation change variation influence in P4L15-19 and supplementary table 2.

R#1: P4L4-6: What labelling algorithm do you use? This bit is unclear and would be difficult to repeat.

A: In order to filter the final products, we group the image in patches with similar values based on the histogram and reject loose regions with area smaller than 1/1000 of the image size.

20 R#1: P4L10-13: It seems like this approach would lump actual physical spatial velocity variability with spurious non-physical velocities. Could this be why your glacier margins/shear zones on Figure 3 are always high uncertainty? There actually is high spatial variability in velocity there, so SNR would be low, but not because of bad data. It seems like this method may overestimate error in some regions.

A: As addressed in the major comments, the signal to noise ratio (SNR) used in this work is a relation between the cross-
25 correlation function peak (Cp) and the average correlation level ($C_L$) on the tracking window used to estimate the velocity (SNR=Cp/$C_L$). The shear margins have high uncertainties because they are challenge areas to track velocity due to the non-uniform flow, lack of stable features and geometry distortion even after the terrain correction.

R#1: P4L17: What is your "airborne estimate of elevation change"? What constitutes an "extreme case" (i.e., what is
30 the magnitude "large" of surface lowering? Needs more detail.

A: Done (P4L22-24).

R#1: P4L15-24: Are these errors systematic or random? Seems like tidal forcing may be random (sometimes surface is higher, other times surface is lower than you think) but thinning would be systematic (always lower than you think).

35 A: The errors are systematic and now we made this clearer in the text.

R#1: P4L26-27: It seems that using one SAR estimate of velocity to check another SAR estimate of velocity is vulnerable to errors that would effect both in the same way. Do you have any known velocities from GPS or optical image correlation to compare against?

40 A: We chose TSX due to its long time series and overlap estimates. The use of GPS velocities to validate our estimates would be a better alternative instead of other SAR estimate, however we don't have any known of overlap measurements.

R#1: P4L26-27: Should also specify that TSX is based off phase change (InSAR) and you are using feature tracking of SAR imagery (amplitude based) if that is indeed the case. This provides more evidence that using it is a robust check on your data because it is a different method.

A: Done (P4L35).

R#1: P5L7-9: How does this "highlight the importance of resolving glacier velocities within their near terminus regions"? And resolve for what purposes?

A: We rephrase the sentence making it clearer (P5L15-17).

10  R#1: P5L25: What station is this 5 year speed change calculated at? Jif? Looks like JI1/2 are relatively stable over this time. What would magnitude be if calculated there?

A: We indicated which point (Jif) we referred to and included the figures that shows it. J1 and J2 have also a negative trend, however gentler ($\sim$40 and $\sim$8 m yr$^{-2}$) due to their distance from the ice front (P5L34-35).

15  R#1: P6L1 - This is interesting that you find different relationship between speedup and retreat at different glaciers. But seems similar to findings by Moon et al., that find some glaciers terminus-forced, others see more land-terminating style of meltwater-forced. If dL = uice −ucalve, then would expect lengthening if calving can't keep up with faster ice flow in summer.

A: Our dataset shows that for this period of time, PG speedup and underwent an ice front advancing.

R#1: P6L20-24 – How could Münchon et al. [2016] have used data from 2016/2017? And seems like comparing apples to oranges to compare speedup calculated over different time spans. That seems to tell more about glacier dynamical change than measurement accuracy.

A: We agree this was poorly worded. We have modified the text to explain more clearly the time period used for the inter-comparison. Moreover, we rewrote the text to specify that we assessed the level of stability and not the accuracy of the dataset.

**FIGURES**

R#1: Figure 1 – difficult to see lines. Please thicken all lines, but especially green. I also think the overlay makes the image harder to interpret. Maybe just overlay on a grayscale hill-shaded DEM that would provide topographic data but not confuse velocity information.

A: Done. However, instead of adding the hill-shaded DEM, we added contour lines with elevation information.

R#1: Figure 2 – This figure would be clearer to me if you labelled the lines in at least panel a.

35  A: Done.

R#1: Figure 5 – Could you show earlier data (from a different source) to put this plot in context of longer-term JI velocity evolution?

A: Due to the focus of the study which is on short-term velocity variations, we prefer to concentrate only on the period after 2009 when sampling is frequent, as we do not aim to analyse long term variations.

R#1: Figure 6 – How are portions of data for fitting red and black lines decided? Don't think either is the 2009-2011 or 2012-2017 fit lines mentioned in the text?

A: The portions are defined by the slowing down period. The fitting lines were mentioned in the text however, it was missing a reference. The reference is included now. (P5L38-40)

**TECHNICAL CORRECTIONS – TYPOGRAPHICAL ERRORS:**

R#1: P1L33: language is a little sloppy/unclear – marine terminating glaciers still have SMB – should specify this 30

A: Done.

R#1: P1L34-35: "erosion of their termini" → replace with "terminus retreat" or "submarine melting"; I can't tell what you mean and "erosion" could be confused for subglacial bedrock lowering.

A: Done.

15  R#1: P1L38: "high frequency variability" → "high spatial variability" if this is what you mean

A: Done.

R#1: P2L8: "Polar Regions" → "polar regions".

A: Done.

R#1: P2L26: "over the past few years" implies the JI speedup is ongoing (which you show it is not).

A: Done.

R#1: P2L33-35: Sentence starting with "Therefore. . ." is confusing. You are saying -10 km2/a is not a big area change?
25  Could just reword this sentence to be positive (e.g., "glacier area has remained relatively constant") instead of negative (e.g., "glacier area has not changed an unusually large amount")

A: Done.

R#1: P3L1: migration of what? Would call this "terminus retreat rate" if that is indeed what it is.

30  A: Done.

R#1: P3L8: Sentence starting with "Although located. . ." could be reworded to use fewer commas.

A: Done.

**J.R. CARR (REFEREE #2)**

R#2: This paper uses new, high-resolution satellite imagery to assess velocity variations on four large Greenlandic outlet glaciers. Overall, it is very well written, clear and topical. It makes a useful contribution to the field and presents interesting results. It also nicely illustrates the usefulness of these datasets. It is concise but addresses the questions it sets out to answer. I have a few minor comments below, which are noting places where things could be clarified or expressed more clearly. Overall, however, I think it is a really good paper in its present form. Nice to see such a well-put together and concise paper.

A: We would like to thank J.R. Carr for the suggestions and positive comments, which have significantly improved the manuscript. We made all the modifications suggested and we addressed the minor comments as follow below.

**MINOR COMMENTS**

R#2: P1L13: Indicate temporal resolution in brackets.

A: Done.

R#2: P1L19: Give date.

A: Done.

R#2: P1L30: Sentence is a bit hard to follow. Consider splitting.

A: We reworded the sentence (P1L30-33).

R#2: P1L34: Specify the time period as 'last years' will date with the paper.

A: Done (P1L34).

R#2: P1L37: There are other references that are relevant here, e.g. Jensen et al., 2016, Carr et al., 2017 (J Glac). There are a couple of places in the intro where only one or two refs are given, but there are clearly more. Please add a selection of relevant ones and add 'e.g.' to indicate awareness of the others you don't list. Also need a couple of references for the statement about glacier specific and climatic controls.

A: Done (P1L39-40).

R#2: P2L3: ...ice sheet dynamics AND ice discharge, and for assessing....

A: Done.

R#2: P2 L14: Why these glaciers. I'd add a sentence of two. This is sort of given in the next sentence, but it feels like it needs a clear justification at this point.

A: Done. We complemented the paragraph on the "Study areas" (P2L22-23)

R#2: P2L33: Seems odd to switch to area change here, after discussing retreat. I know this paper does look at area change, so I'd work on improving the flow of the argument here.

A: We changed the linking word, showing additional information (P2L36).

R#2: P3L21: Is this the maximum data range? Worth stating for clarity.

A: Done (P3L22-24).

R#2: P3L26: Might be useful to have graph showing image availability for each glacier over time. E.g. you could have a bar graph, with number of images on the Y axis, then time on the X axis, at monthly intervals. You'd then have a different coloured bar for each of the four glaciers. I'm suggesting this so that the reader can get a better handle on how these 187 velocity maps are distributed over the glaciers and over time.

A: Done. We added the figure as a supplementary figure (Figure S2).

R#2: P3L40: Why was this value used?

A: This means values under 5 % from the maximum normalised cross-correlation peak were rejected (P4L1).

R#2: P4L2: Why was this spacing used? Was it because of the GIMP resolution?

A: We use the GIMP for the terrain correction, which the spatial resolution is 90m.

R#2: P4L14: Please state what these higher errors were in a separate sentence.

A: Done.

R#2: P4L43: What period were these means taken over? Or do you mean the maximum value of the means across the whole glacier? Generally this is done well, but make sure you give the time period / spatial extent of your averages, as it's sometimes hard to follow which average/period you are discussing.

A: We meant the mean speeds along the stacked dataset (P5L7).

R#2: P4L44: Do you mean velocity MAXIMA (not magnitude)?

A: Done.

R#2: P5: On this page, there are several points were it would be useful to refer back to the relevant figures, e.g. L2, L24, L26. Please update throughout, as it helps guide the reader quickly to the relevant figures.

A: Done.

R#2: P5L2: I'm not clear what '46 km extension' refers to. Please revise this description so it's clear.

A: Done (P5L11).

R#2: P5L7: I find this hard to follow. Please re-phrase.

A: Done (P5L15-16).

R#2: P5L12: ...scale variability, which continues....

A: Done.

R#2: P5L14: Helpful to repeat the time period these values relate to.

A: Done.

R#2: P5L34: starts IN early 2015.

A: Done.

R#2: P6L10: Please state whether or not this relationship is statistically significant, then have the explanation of why it is not significant (i.e. lack of data).

A: We explain it is difficult to assess due to the limited dataset (P6L26).

R#2: P6L27: I think it would help to add a summary sentence or two. I'd definitely add one to sum up the key message of this paragraph (i.e. your data agree pretty well), and maybe add another, more general summary sentence to reflect on the usefulness of the data for this purpose. At the moment, it feels like the paper ends abruptly, even if you do say this in the conclusion.

A: Done (P6-43-P7L2).

R#2: P6L31: Be specific about what you mean by 'important', I.e. high flow, large discharge.

A: Done (P7L3).

R#2: P6L32: I'd just give the date, as 'the present day' dates.

A: Done (P7L6).

**FIGURES**

R#2: Figure 1 & 3: The labels on these maps need to be much bigger, especially the locations of the extracted velocities. I find it really difficult to see these, but they're important for the context of the paper. I also think the three Greenland overviews are too small and don't work. Instead, please add one Greenland overview, with the sites marked, but which is a reasonable size. It's a shame to have nice figures like these when the reader can't read them properly.

A: Done.

R#2: Figure 2: As with the other figures, this needs to be bigger, especially the text, as I can barely read it, especially the axes labels.

A: Done.

R#2:Figure 3: The grounding line needs to be much more obvious, as do the lines for the termini. I really struggle to see them. Same for figure 1 Particularly for A), it would be useful to have a land mask to orientate people.

A: Done.

R#2:Figure 4: make the line stronger ad markers larger. Add the p-value and R2 for this regression.

A: We added R2 values to the regression.

R#2:Figure 5: The text is much easier to read on this, but some of the points are hard to see, e.g. the green dots in A. I know it's difficult given the data density, but think how you can back the points easier to see throughout this figure (e.g. through increasing size or changing the colours).

A: Done.

R#2:Figure 6: Add p-values for the regression lines.

A: We include the correlation coefficients to the regression lines.

R#2:Figure 7: Might be clearer with slightly large points.

A: Done.

[revised manuscript text omitted]

-

---

## Author Comment (AC2) · 12 May 2018

Thank you for your review, please find attached our response to all referee comments.

Please also note the supplement to this comment:
https://www.the-cryosphere-discuss.net/tc-2017-251/tc-2017-251-AC2-supplement.pdf

---

## Author Response (AR2)

30th May 2018

Centre for Polar Observation and Modelling

School of Earth and Environment

University of Leeds

5    LS2-9JT, Leeds - UK

Dear Prof. Vieli,

**"Ice velocity of Jakobshavn Isbræ, Petermann Glacier, Nioghalvfjerdsfjorden and Zachariæ Isstrøm, 2015-2017, from Sentinel 1-a/b SAR imagery"**

We are grateful for all the comments provided by yourself. Please find attached the revised manuscript that

15    addresses each of the minor points raised.

Yours faithfully

Adriano Lemos

20    (corresponding author)

[revised manuscript text omitted]